# Cerebrospinal Fluid Mitochondrial DNA in Rapid and Slow Progressive Forms of Alzheimer’s Disease

**DOI:** 10.3390/ijms21176298

**Published:** 2020-08-31

**Authors:** Petar Podlesniy, Franc Llorens, Margalida Puigròs, Nuria Serra, Diego Sepúlveda-Falla, Christian Schmidt, Peter Hermann, Inga Zerr, Ramon Trullas

**Affiliations:** 1Neurobiology Unit, Institut d’Investigacions Biomèdiques de Barcelona (CSIC), 08036 Barcelona, Spain; ppodlesniy@gmail.com (P.P.); puigrosserra.m@gmail.com (M.P.); serranuri@gmail.com (N.S.); 2Centro de Investigación Biomédica en Red sobre Enfermedades Neurodegenerativas (CIBERNED), 08036 Barcelona, Spain; franc.llorens@gmail.com; 3Bellvitge Biomedical Research Institute (IBIDELL), 08908 L’Hospitalet de Llobregat, Spain; 4Department of Neurology, Clinical Dementia Center, University Medical School, Georg-August University, 37075 Göttingen, Germany; Christian.Schmidt@medizin.uni-goettingen.de (C.S.); peter.hermann@med.uni-goettingen.de (P.H.); IngaZerr@med.uni-goettingen.de (I.Z.); 5Institute of Neuropathology, University Medical Center Hamburg-Eppendorf, 20251 Hamburg, Germany; d.sepulveda-falla@uke.uni-hamburg.de; 6German Center for Neurodegenerative Diseases (DZNE), 37075 Göttingen, Germany; 7Institut d’Investigacions Biomèdiques August Pi i Sunyer (IDIBAPS), 08036 Barcelona, Spain

**Keywords:** mitochondrial DNA, cerebrospinal fluid, Alzheimer’s disease, biomarker, digital PCR

## Abstract

Alzheimer’s type dementia (AD) exhibits clinical heterogeneity, as well as differences in disease progression, as a subset of patients with a clinical diagnosis of AD progresses more rapidly (rpAD) than the typical AD of slow progression (spAD). Previous findings indicate that low cerebrospinal fluid (CSF) content of cell-free mitochondrial DNA (cf-mtDNA) precedes clinical signs of AD. We have now investigated the relationship between cf-mtDNA and other biomarkers of AD to determine whether a particular biomarker profile underlies the different rates of AD progression. We measured the content of cf-mtDNA, beta-amyloid peptide 1–42 (Aβ), total tau protein (t-tau) and phosphorylated tau (p-tau) in the CSF from a cohort of 95 subjects consisting of 49 controls with a neurologic disorder without dementia, 30 patients with a clinical diagnosis of spAD and 16 patients with rpAD. We found that 37% of controls met at least one AD biomarker criteria, while 53% and 44% of subjects with spAD and rpAD, respectively, did not fulfill the two core AD biomarker criteria: high t-tau and low Aβ in CSF. In the whole cohort, patients with spAD, but not with rpAD, showed a statistically significant 44% decrease of cf-mtDNA in CSF compared to control. When the cohort included only subjects selected by Aβ and t-tau biomarker criteria, the spAD group showed a larger decrease of cf-mtDNA (69%), whereas in the rpAD group cf-mtDNA levels remained unaltered. In the whole cohort, the CSF levels of cf-mtDNA correlated positively with Aβ and negatively with p-tau. Moreover, the ratio between cf-mtDNA and p-tau increased the sensitivity and specificity of spAD diagnosis up to 93% and 94%, respectively, in the biomarker-selected cohort. These results show that the content of cf-mtDNA in CSF correlates with the earliest pathological markers of the disease, Aβ and p-tau, but not with the marker of neuronal damage t-tau. Moreover, these findings confirm that low CSF content of cf-mtDNA is a biomarker for the early detection of AD and support the hypothesis that low cf-mtDNA, together with low Aβ and high p-tau, constitute a distinctive CSF biomarker profile that differentiates spAD from other neurological disorders.

## 1. Introduction

Alzheimer’s disease (AD) is a progressive neurodegenerative disorder that exhibits marked heterogeneity regarding its etiology and age of clinical onset. Moreover, different rates of progression have been observed among patients with clinical signs of AD type dementia, suggesting the existence of distinct disease subtypes [1,2,3,4]. A commonly used classification of AD subtypes based on rate of cognitive decline defines rapid progressive AD (rpAD) as a decrease of six points per year in the score of the mini-mental state examination test [5,6], although alternative definitions for rpAD have also been proposed [7,8,9]. Converging lines of evidence indicate that differences in the speed and slope of cognitive decline correlate with neuropathological hallmarks of AD, with clinical features of brain atrophy or with genetic factors [1,4,5,10,11]. However, the pathophysiological mechanisms that underlie the differences in progression rate of cognitive decline in AD type dementia are still unknown.

The diagnosis of possible or probable AD utilizes clinical criteria with increasing support from neuroimaging and cerebrospinal fluid (CSF) core AD related biomarkers such as total-tau (t-tau), phospho-tau (p-tau) and beta-amyloid 1-42 (Aβ). These CSF biomarkers are able to predict the rate of cognitive decline in AD [12,13,14]. Additionally, CSF t-tau, and especially Aβ are altered years before the appearance of clinical signs [14]. However, in relationship with AD progression, contradictory data have been reported regarding differences in CSF profiles between slow (spAD) and rapid (rpAD) progression of the disease [15]. The diagnosis of rpAD may be challenging due to partial clinical overlap with other rapidly progressive neurodegenerative dementias such as Creutzfeldt-Jakob disease [15,16]. Despite the limited information about the molecular basis of rpAD, the rate of cognitive decline has been associated with the presence of diverse structural assemblies of Aβ [17,18] and with a proteomic content in amyloid plaques different that found in spAD [19]. However, whether these differences result in a unique biomarker profile that distinguishes spAD from rpAD at the early stages of cognitive decline has not been explored in detail. 

We previously found that patients with spAD, and no other dementias, which have the two core AD related biomarkers low Aβ and high t-tau in CSF, exhibited low CSF content of cell-free mitochondrial DNA (cf-mtDNA) compared to control subjects without any AD-related biomarker or known genetic AD risk factor such as ApoE4 [20,21]. In addition, subjects carrying pathogenic mutations in the PSEN1 gene, but without clinical signs of AD or altered CSF biomarkers, also exhibited low content of CSF cf-mtDNA [20]. Overall, these results suggested that a decrease in the CSF content of cf-mtDNA precedes the clinical signs and differentiates the subgroup of spAD type dementia. As a continuation of these previous studies, we have now investigated the relationship between the CSF content of mtDNA and the other core AD biomarkers to assess whether there is a unique molecular profile that may differentiate the two forms of AD disease progression.

## 2. Results

### 2.1. Characteristics of Study Cohort: AD Biomarkers in Patients and Neurological Disease Controls

Table 1 shows the characteristics and CSF levels of core AD-related biomarkers in the study cohort distributed into neurological disease without dementia (ND-Control) and AD patient groups and stratified according to the presence/absence of AD biomarker in the CSF and disease progression. Overall, the AD patient group showed statistically significant increases of 101% in t-tau, a 115% in p-tau and a decrease of 31% in Aβ CSF levels over the ND-Control group. In addition, the AD patient group, all together, showed a decrease of 31% in CSF levels of cf-mtDNA-85 compared to the ND-Control Group (Table 1).

Among the total 49 control patients diagnosed with diverse neurological diseases but without clinical signs of AD type dementia, 18 patients (37%) presented at least one core biomarker of AD (Aβ ≤ 450 or t-tau ≥ 450): five controls with only high t-tau, 10 controls with low Aβ, and three controls with both biomarkers. On the other hand, among the total 46 patients with clinical signs of AD dementia, 23 (50%) did not accomplish the criteria of presence of two core AD biomarkers: high t-tau and low Aβ in CSF. Of these clinical AD cases without two core AD biomarkers, 16 patients (70%) displayed spAD while seven patients (30%) exhibited rpAD. No significant differences were observed in age and gender representation between groups.

### 2.2. cf-mtDNA in AD Disease Progression

First, we analyzed the CSF content of cf-mtDNA in relationship to clinical diagnosis, independently of biomarker criteria, in all cohort subjects distributed in three different groups: control subjects diagnosed with nonprimarily neurodegenerative neurological and psychiatric diseases without dementia (ND-Controls); patients with clinical signs of AD type dementia of slow progression (spAD), and patients with clinical signs of AD of rapid progression (rpAD). In this whole cohort, patients with spAD showed a statistically significant decrease of 44% in CSF levels of cf-mtDNA-85 over the ND-Control Group (ND-Ctrl = 62.0 cf-mtDNA-85 copies/ul CSF, 95% CI 47–77, *n* = 49; spAD = 34.6 cf-mtDNA-85 copies/ul CSF, 95% CI 23–46, *n* = 30, *p* = 0.02). In contrast, the levels of CSF cf-mtDNA-85 from patients with rpAD were not statistically significantly different from those found in ND-Controls (rpAD = 57.6 cf-mtDNA-85 copies/ul CSF, 95% CI 30-85, *n* = 16, *p* = 0.9), (Figure 1A).

Next, we analyzed the CSF content of cf-mtDNA in groups formed according to the AD biomarker profile of the subjects. In the ND-Control group we included only those subjects who were biomarker negative and did not show any of the two core CSF biomarkers of AD (Aβ ≤ 450 or t-tau ≥ 450), and in the spAD and rpAD groups we included only those patients who were biomarker positive and had both AD biomarkers. In this biomarker-selected cohort, the decrease in CSF content of cf-mtDNA in patients with spAD was of higher magnitude than that observed in the unselected whole cohort. Thus, cf-mtDNA-85 in the spAD group showed a statistically significantly reduction of 69% over the ND-Control Group (ND-Ctrl = 69.4 cf-mtDNA-85 copies/uL CSF, 95% CI 48–91, *n* = 31; spAD = 21.7 cf-mtDNA-85 copies/uL CSF, 95% CI 15–29, *n* = 14, *p* = 0.003). Similar to what we observed previously in the whole cohort, in this biomarker-positive cohort the levels of CSF cf-mtDNA-85 from patients with rpAD were not statistically significantly different from those found in ND-Controls (rpAD = 54.8 cf-mtDNA-85 copies/uL CSF, 95% CI 14–96, *n* = 9, *p* = 0.8), (Figure 1B).

### 2.3. Relationship between cf-mtDNA-85 and Core CSF Biomarkers of AD

Subsequently, we studied the relationship between CSF content of cf-mtDNA with other AD biomarkers in the entire cohort, including all subjects, independently of biomarker or clinical criteria. Regression analyses revealed a statistically significant positive correlation between CSF content of cf-mtDNA-85 and Aβ levels (r = 0.38, 95% CI 0.19–0.54, *p* = 0.0001, *n* = 95, Figure 2A) and a statistically significant negative correlation between cf-mtDNA-85 and levels of p-tau in CSF (r = −0.21, 95% CI −0.40–0.01, *p* = 0.041, *n* = 95, Figure 2E). In contrast, we did not find significant relationship between cf-mtDNA-85 and t-tau levels in CSF (r = 0.03, *p* = 0.78, *n* = 95, Figure 2C), as expected, as t-tau levels are affected in later stages of disease progression.

To confirm the predictive ability of cf-mtDNA-85, we analyzed the CSF content of cf-mtDNA-85 in the entire cohort of subjects segregated in two groups based on the cut-off value of each AD core biomarker. In conformity with the relationship reported above, subjects with CSF levels of Aβ equal or below the cut-off value of 450 pg/mL exhibited statistically significant low CSF content of cf-mtDNA-85 compared to the group with Aβ beyond the cut-off value. (Aβ > 450 = 67.7 cf-mtDNA-85 copies/uL CSF, 95% CI 53–83, *n* = 50; Aβ ≤ 450 = 35.8 cf-mtDNA-85 copies/uL CSF, 95% CI 26–46, *n* = 45, *p* = 0.002, Figure 2B). In addition, subjects with CSF levels of p-tau equal or above the cut-off value of 61 pg/mL showed statistically significant low CSF content of cf-mtDNA-85 compared to the group with p-tau below the cut-off value. (p-tau < 61 = 63.7 cf-mtDNA-85 copies/uL CSF, 95% CI 48–80, *n* = 48; p-tau ≥ 61 = 41.3 cf-mtDNA-85 copies/uL CSF, 95% CI 31–52, *n* = 47, *p* = 0.03, Figure 2F). In contrast, segregation of the cohort subjects by their CSF level of t-tau did not result in statistically significant different content of cf-mtDNA-85 (Figure 2D).

### 2.4. Ratio of cf-mtDNA-85 over p-tau in AD Progression

Based on the opposite relationship between cf-mtDNA-85 and p-tau, we evaluated the specificity and sensitivity of the ratio between cf-mtDNA-85 over p-tau to discriminate ND-control subjects from patients with AD. In the whole cohort, the cf-mtDNA-85/p-tau ratio was statistically significantly low in patients with spAD and rpAD by 33% and 21%, respectively (ND-Ctrl = 1.04, 95% CI 0.97–1.11, *n* = 49; spAD = 0.70, 95% CI 0.61–0.79, *n* = 30, *p* < 0.0001; rpAD = 0.82, 95% CI 0.67–0.96, *n* = 16, *p* = 0.04) (Figure 3A). Receiving operating curve analyses (ROC) in the whole cohort revealed that using a cutoff value of <0.885 the cf-mtDNA-85/p-tau ratio distinguishes the diagnosis of AD (spAD and rpAD combined) from ND-controls with a sensitivity of 70%, specificity of 69% and area under the ROC of 0.791 (95% CI = 0.70–0.88, *p* < 0.0001) (Figure 3B). Similar ROC values were obtained in distinguishing the diagnosis of rpAD alone with a sensitivity of 77%, specificity of 69%, and area under the ROC of 0.839 (95% CI = 0.75–0.93, *p* < 0.0001) (Figure 3C).

When control and patient groups were selected by biomarker criteria, the specificity and sensitivity values improved. In the biomarker-selected groups, the cf-mtDNA-85/p-tau ratio was statistically significantly reduced in patients with spAD and rpAD by 44% and 34%, respectively, compared to the ND-Control group (ND-Ctrl = 1.11, 95% CI 1.02–1.21, *n* = 31; spAD = 0.62, 95% CI 0.52–0.72, *n* = 14, *p* < 0.0001; rpAD = 0.73, 95% CI 0.55–0.91, *n* = 9, *p* = 0.007) (Figure 3D). ROC analyses in these biomarker-selected groups showed that the cf-mtDNA-85/p-tau ratio distinguishes the diagnosis of AD from ND-Controls with a sensitivity of 83%, specificity of 81% and area under the ROC of 0.920 (95% CI = 0.85–0.99, *p* < 0.0001) (Figure 3E). Likewise, improved ROC values were obtained in distinguishing the diagnosis of spAD alone with a sensitivity of 93%, specificity of 94%, and area under the ROC of 0.972 (95% CI = 0.93–1.01, *p* < 0.0001) (Figure 3F).

## 3. Discussion

Previous studies reported low cf-mtDNA content in the CSF from patients with AD and unveiled the potential of cf-mtDNA quantification as a potential biomarker for the early detection of preclinical AD [20,21]. Nonetheless, another study in a wide series of patients and controls found a high interindividual variability in CSF cf-mtDNA content [22]. Given the high molecular and clinical heterogeneity of AD, we investigated whether the content of cf-mtDNA in CSF was associated with core AD biomarkers and influenced by disease progression rate; with the aim to determine whether a particular cf-mtDNA profile underlies the biomarker variability found in the different forms of AD.

First, the present results confirm our previous observations showing that the CSF of AD patients contains statistically significantly lower concentrations of cf-mtDNA compared to controls, adding another line of evidence for the association between cf-mtDNA and AD pathology. Secondly, in the present study we observed that cf-mtDNA in CSF correlates positively with Aβ and negatively with p-tau, but not with t-tau. Indeed, cf-mtDNA concentrations were statistically significantly lower in cases with pathogenic CSF concentration of Aβ (<450 pg/mL) and p-tau (>61 pg/mL) compared to those with normal CSF values of both biomarkers. On the one hand, low CSF content of Aβ is the core AD biomarker that appears earliest in the CSF during the clinical course of the disease [18]. In this regard, it is well stablished that amyloid accumulation precedes alterations in brain structure, cognition and clinical signs of AD, which may result in changes in the CSF content of Aβ and p-tau [23,24]. Consequently, the positive relationship between cf-mtDNA and Aβ levels in the CSF found in the present study is consistent with our previous report showing low CSF cf-mtDNA content in both presymptomatic carriers of pathogenic PSEN1 mutations and asymptomatic patients at risk of AD [20]. 

On the other hand, it has also been shown that the CSF content of p-tau, a marker of tau-related pathology in brain tissue, is already altered at preclinical AD stages. While some studies suggest that changes in the content of p-tau in CSF occur later in the disease process than those of Aβ, there is strong evidence showing that the CSF levels of Aβ, but not those of t-tau, are altered already five to ten years before the appearance of the clinical signs of Alzheimer’s dementia. [25]. Furthermore, other studies postulate that CSF levels of p-tau may change as soon as Aβ levels in preclinical AD [26]. Hence, the negative correlation found in the present study between cf-mtDNA and p-tau provides further evidence to support the hypothesis that a decrease in the CSF content of cf-mtDNA is a pathophysiological biomarker of AD. 

Moreover, the positive correlation between cf-mtDNA and Aβ, together with the lack of association between cf-mtDNA and t-tau, a surrogate marker of neuro-axonal damage, indicates that a decrease in cf-mtDNA is a marker of AD pathogenesis independent of the degree of neuronal damage. These results are also in line with the observation that cf-mtDNA is not altered in Creutzfeldt-Jakob disease (CJD), a rapidly progressive neurodegenerative dementia characterized by the presence of massive synaptic and neuronal damage and with the absence of correlation between the CSF content of cf-mtDNA and t-tau in CJD [21]. Altogether, the present results provide further evidence for the hypothesis that alterations in CSF content of cf-mtDNA precede the neurodegenerative process that leads to the emergence of clinical signs of dementia. 

Notably, in the present study, we found that 37% of nonAD dementia controls had at least one AD biomarker. Furthermore, 53% and 44% of subjects with clinical signs of spAD and rpAD, respectively, did not fulfill the two core AD biomarker criteria of high t-tau and low Aβ in CSF. These observations are consistent with previous evidence indicating that purely clinical diagnostic criteria for AD have poor accuracy, with specificity and sensitivity values between 70–80% when compared with neuropathology [18,27]. This is likely due to the dynamic nature of biomarkers: those linked to the pathophysiology of the disease emerging in the prodromal stage, while those more associated with disease progression increasing at later disease stages. More longitudinal studies are necessary to stablish the dynamics of CSF biomarkers during the evolution of different clinical dementia subtypes. The presence of at least one AD biomarker in subjects from the control group (21% of controls had low Aβ, 10% had high t-tau, and 6% both), together with the absence of one of these biomarkers in subjects with clinical signs of dementia, underscores the need to use a combined biomarker profile to improve accuracy of the diagnosis of AD. 

According to this necessity, and based on the relationship we found between cf-mtDNA and p-tau, we evaluated the accuracy of their ratio to discriminate between ND-Control subjects and AD. Interestingly, the sensitivity and specificity of the cf-mtDNA/p-tau ratio to distinguish the diagnosis of AD was markedly higher in the core AD biomarker-selected cohort than in the unselected whole cohort (83% and 81% in biomarker-selected cohort vs 70% and 69% in whole cohort). The sensitivity and specificity values increased further, up to 93% and 94%, respectively, when the ratio between CSF content of cf-mtDNA-85 and p-tau was evaluated to discriminate the diagnosis of spAD without rpAD in the biomarker-selected group (Figure 3F). These results support the hypothesis that the ratio cf-mtDNA-85/p-tau constitutes a distinctive CSF biomarker profile that differentiates slow AD progression from patients with other neurodegenerative diseases. Moreover, the present results indicate that low content of CSF mtDNA defines a group of spAD cases characterized by a biomarker profile of low levels of Aβ and high levels p-tau in CSF, and this biomarker profile may serve to stratify a subgroup of AD type dementias different from other dementia subtypes.

With regard to the difference in disease progression, in contrast with spAD, the CSF content of cf-mtDNA was not statistically significantly different in rpAD cases compared to controls, both in the whole cohort and in the cohort of patients selected by the core AD biomarker criteria (Aβ and t-tau). Despite mean cf-mtDNA concentrations being higher in rpAD compared to spAD, our results do not support a role for mtDNA in the different rate of AD clinical progression. However, additional studies are needed to ascertain why CSF values of cf-mtDNA are higher in some rpAD patients.

One significant limitation of the present study that might influence the lack of statistically significant difference between spAD and rpAD groups is the small size of the rpAD group, both in the whole cohort and in the biomarker-selected cohort. Nonetheless, the group with rpAD exhibits two different types of patients, one with high and the other with low amount of CSF mtDNA suggesting that the molecular mechanism involved in the rapid course of AD neurodegeneration is unrelated to cf-mtDNA. Further studies analyzing the neuropathological differences between these two types of patients with rpAD are necessary to determine whether they are affected by comorbidity with other neurological diseases.

## 4. Materials and Methods 

### 4.1. Subjects

The study included 95 CSF samples from patients diagnosed with neurological diseases without dementia (ND-Controls, *n* = 49) and Alzheimer’s disease (AD, *n* = 46). Control cases included nonprimarily neurodegenerative neurological and psychiatric diseases (ND-Controls) and encompassed the following diagnostic groups: psychosis, depression, polyneuropathy, bipolar disorder, schizophrenia, postencephalitic epilepsia, meningitis, headache, vertigo, paraneoplasia, inflammatory and autoimmune disease and pain syndromes. ND-Controls were diagnosed according to acknowledged standard neurologic clinical and paraclinical findings assessment based on the International Classification of Diseases and Related Health Problems, Tenth Edition definitions. Only patients with either clinically or pathologically defined diagnoses were included. ND-Controls did not present cognitive impairment or dementia at the time of sampling.

AD cases were collected in the framework of the rpAD study, an ongoing prospective study at the Clinical Dementia Center Göttingen (Germany) [28,29]. AD cases were diagnosed according to the National Institute on Aging–Alzheimer’s Association workgroups criteria [30] and further stratified in slow progressive AD (sAD) and rapidly progressive AD (rpAD). rpAD was defined by a velocity of cognitive decline of >6 points/year on the Mini-Mental Status Examination scale (MMSE) as proposed by Schmidt et al. [5]. All other patients diagnosed with probable AD with MMSE cognitive decline equal or lower than six points per year were considered with slow progression AD (spAD). Velocity of decline was calculated using linear regression (least square method) in accordance with Villemagne et al. [31]. The study was conducted according to the revised Declaration of Helsinki and Good Clinical Practice guidelines. Ethics approval was obtained from the local Ethics Committee of the University of Göttingen (Number 11/11/93 (+Amendments) and Number 9/6/08). All study participants or their legal guardians provided written informed consent.

### 4.2. AD Biomarker Analysis in CSF Samples

The amount of t-tau was quantitatively analyzed using the commercially available enzyme-linked immunosorbent assay (ELISA) INNOTESThTAU-Ag kit from Fujirebio. The amount of p-tau phosphorylated at Thr181 was measured with the INNOTEST PHOSPHO-TAU(181P) ELISA kit from Fujirebio. The amount of Aβ was quantified using the INNOTEST ß-AMYLOID (1–42) ELISA kit from Fujirebio. AD cut-offs for t-tau and Aβ were 450 pg/mL according to data from the rpAD study.

### 4.3. Droplet Digital PCR

The content of cf-mtDNA was measured directly in CSF without nucleic acid extraction by droplet digital PCR (dPCR) using Bio-Rad QX200 platform (Bio-Rad Laboratories, Hercules, CA, USA) as previously described [21,32]. The PCR reaction was performed in ddPCR Supermix for Probes (No dUTP) (Ref: 186-3023, Bio-Rad) with the mtDNA-85 primer pair that targets a region of the cytochrome B gene corresponding to bases 14848-14932 of the human mtDNA Cambridge reference sequence NC_012920.1 and producing an amplicon of 85 base pair length. The sequences of the primers and hydrolysis probe are:

Forward: mtDNA-85F (5′-CTCACTCCTTGGCGCCTGCC-3′); 

Reverse: mtDNA-85R (5′-GGCGGTTGAGGCGTCTGGTG-3′); 

Probe: (FAM) CCTCCAAATCACCACAGGACTATTCCTAGCCATGCA (BHQ1).

Each CSF sample was also tested for the presence of nuclear DNA to discard the possibility of contamination with mtDNA from lysed cells. Nuclear DNA was assessed simultaneously with mtDNA in the same CSF sample by measuring the single copy nuclear gene BAX in a multiplex assay. Forward and reverse primers and hydrolysis probe BAX sequences are: 

Forward: BAX-103F, (5′-TTCATCCAGGATCGAGCAGG-3′);

Reverse: BAX-103R, (5′-TGAGACACTCGCTCAGCTTC-3′);

Probe: HEX-BAX-103P, (HEX)-5′-CCCGAGCTGGCCCTGGACCCGGT-3′-BHQ1. 

Complete details of multiplex cf-mtDNA and BAX assay optimization in CSF using the mtDNA-85 and BAX-103 hydrolysis probes have been previously described [32]. Following dMIQE guidelines [33,34], characterization studies showed that the optimum temperature to reach the maximum separation between positive and negative droplets in our reaction conditions for both mtDNA-85 and BAX-103 amplicons is 60 °C. Accordingly, the PCR amplification was performed using the following thermal profile: 95 °C 10 min; (94 °C 30 s; 60 °C 1min) 40 repeats; 98 °C 10min; 4 °C infinite, in the C1000 Touch Deep Well Thermal Cycler (Bio-Rad).The assay was performed directly in unpurified CSF samples. We found that up to three freeze-thaw cycles did not significantly modify the measured concentration of mtDNA in CSF by dPCR. Previous characterization studies showed that the volume of unpurified CSF sample that can be added to the dPCR reaction without inhibition of the amplification due to factors present in CSF could be up to 6 uL in a total reaction volume of 20 uL. In the present study, we used 4.5 uL of CSF in a 20 uL dPCR reaction, which provides the highest signal with the lowest possibility of amplification inhibition. To test the possibility that different individual CSF samples might have different amounts of inhibitory factors, we added known amounts of mtDNA and BAX to CSF samples with different concentrations of Aβ and tau. Despite a difference in approximately two orders of magnitude in the concentration of these two proteins, and possibly of many other proteins, we found no significant differences in the expected amount of mtDNA-85 and BAX-1003 spiked in these samples. Measurement of mtDNA or BAX in a sample volume of 4.5 uL of CSF was linear up to 3000 copies/uL of CSF, indicating that in the conditions of our dPCR reaction there is no influence of the inhibitory factors present in the CSF. Moreover, the presence of inhibition in the dPCR reaction results in a decrease in the fluorescence amplitude of the positive droplets. In the present study, no significant difference was observed in fluorescence amplitude of positive droplets between samples, neither within nor between groups, confirming the absence of differential inhibition in our dPCR reaction conditions. The average amount of total droplets obtained in our assays was 16,500 ± 126, with a relative uncertainty of cf-mtDNA positive droplets below 10% due to the low amount of cf-mtDNA found in CSF samples. The limit of detection was close to the theoretical limit of three copies of target per reaction, corresponding to less than one copy of target per microliter of CSF with a signal above the 95% confidence interval of average noise in nontemplate controls. Agarose gel electrophoresis analyses confirmed that the dPCR reaction conditions used in our experiments produced single amplicons of the corresponding size, 103 and 85 base pairs for Bax-103 and mtDNA-85 primer combinations, respectively. Samples with a number of BAX copies > 1/uL CSF were not included in analyses. Reactions were performed in triplicate in 96 well plates. Each analysis plate included a balanced number of samples from all groups and nontemplate controls.

### 4.4. Statistical Analysis

We used GraphPad Prism software v7 for statistical tests. We performed receiving operating curve analyses to evaluate the diagnostic sensitivity and specificity of cf-mtDNA content in CSF in classifying AD patients versus ND-Control subjects. Values were expressed as mean and 95% confidence interval CI. We balanced samples from all groups in all dPCR assays. No values were removed for statistical purposes. For linear regression and receiving operating curve (ROC) analyses, values were transformed to natural logarithm. Statistical analyses were performed using Kruskal-Wallis one-way analysis of variance with Dunn’s multiple comparisons post hoc tests. Comparison between two groups was performed with two-tailed Mann-Whitney U-tests or with two-tailed unpaired Student’s t-tests depending on whether data passed D’Agostino & Pearson normality test. Differences were considered statistically significant at a value of *p* < 0.05.

## 5. Conclusions

The present results indicate that the content of cf-mtDNA in CSF correlates with the pathological markers of the disease Aβ and p-tau, but not with the marker of neuronal damage t-tau. In addition, the present results support the hypothesis that low cf-mtDNA, together with low Aβ and high p-tau, constitute a distinctive CSF biomarker profile that differentiates slow AD progression from patients with other neurodegenerative diseases.

## Figures and Tables

**Figure 1 ijms-21-06298-f001:**
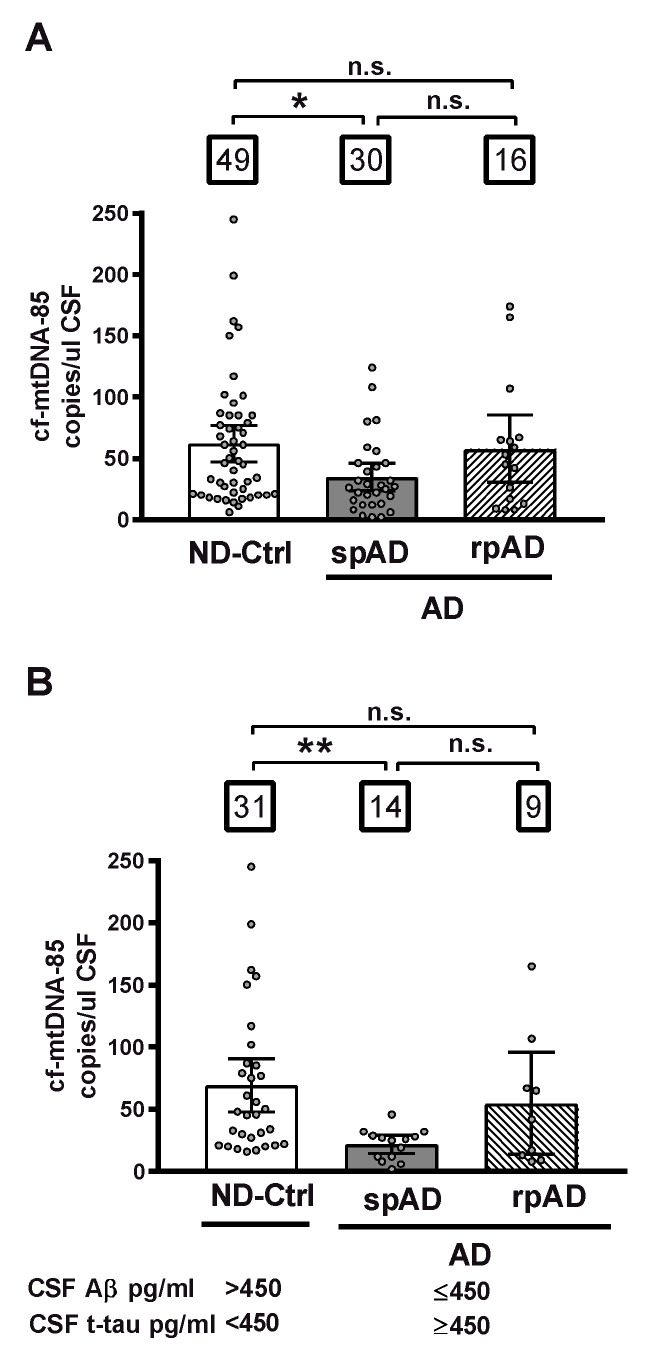
CSF content of cf-mtDNA-85 in AD disease progression. The absolute number cf-mtDNA-85 copies was measured directly in CSF without nucleic acid extraction by droplet digital PCR (dPCR) in the presence of a hydrolysis probe using the mtDNA-85 primer pair, which targets a region of the cytochrome B gene producing an amplicon of 85 base pairs length. The CSF content of cf-mtDNA-85 from all subjects of the whole cohort is shown in (**A**) and from subjects selected using the indicated cut-off values of the core AD biomarkers Aβ and t-tau is shown in (**B**). Subjects were distributed in three different groups: ND-Ctrl, control group composed of patients with neurological diseases and with no clinical signs of AD type dementia; spAD, group composed of patients with clinical signs of AD type dementia of slow progression; rpAD: group composed of patients with clinical signs of AD of rapid progression. Numbers within squares show the number of patients in each group. Dots represent individual values. Bars represent mean ± 95% CI. ** Statistically significantly different *p* < 0.01, * statistically significantly different *p* < 0.05; n.s = statistically nonsignificantly different.

**Figure 2 ijms-21-06298-f002:**
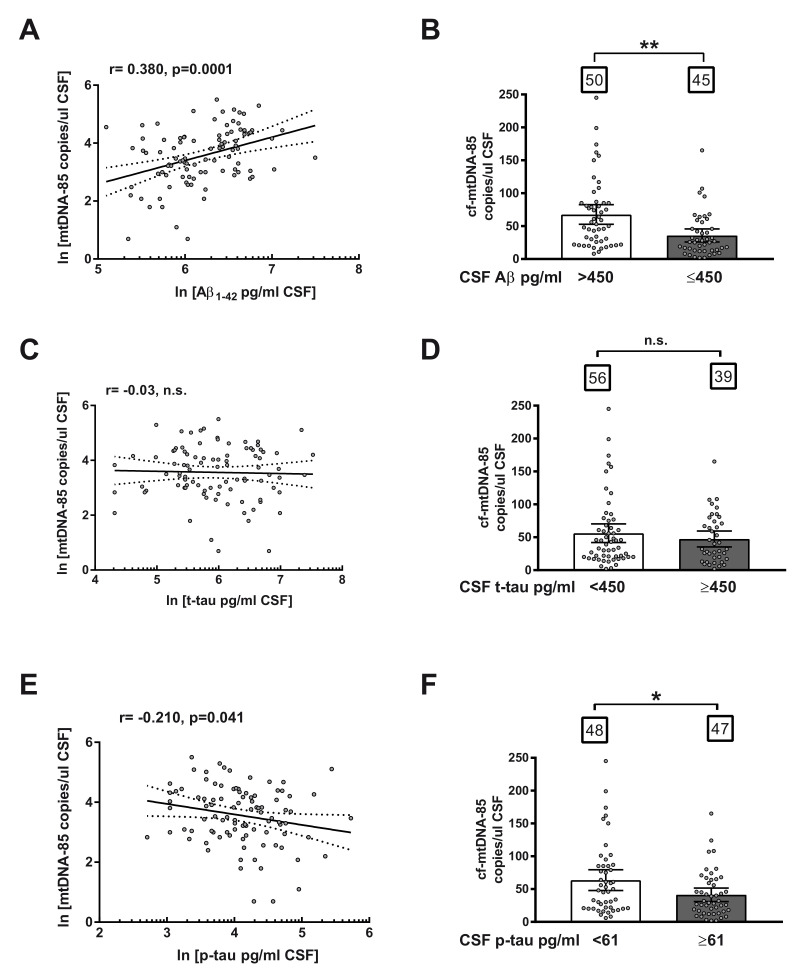
Relationship between *cf-mtDNA-85 and CSF biomarkers of AD*. (**A**,**C**,**E**): linear regression graphs and Pearson correlation (r) between CSF content values of cf-mtDNA-85 and Aβ (**A**), t-tau (**C**) or p-tau (**E**). Dotted lines represent 95% CI. Values are from all subjects of the entire cohort (*n* = 95). Dots represent individual values. All values were transformed to natural logarithm. *n*.s = nonsignificant. (**B**,**D**,**F**): bar graphs showing the CSF cf-mtDNA-85 content from subjects of the entire cohort segregated in two groups according to the cut-off value indicated for Aβ (**B**), t-tau (**D**) or p-tau (**F**). Numbers within squares show the number of patients in each group. Dots represent individual values. Bars represent mean ± 95% CI. ** Statistically significantly different, *p* < 0.01; * statistically significantly different, *p* < 0.05. n.s = statistically nonsignificantly different.

**Figure 3 ijms-21-06298-f003:**
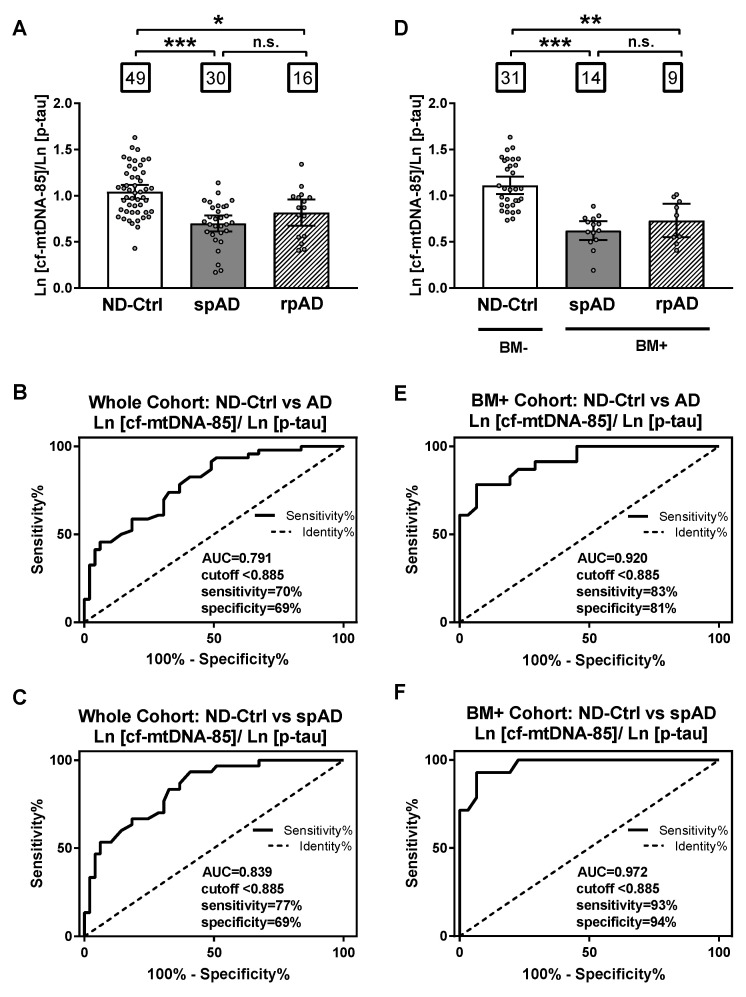
Sensitivity and specificity *of cf-mtDNA-85/p-tau ratio*. (**A**,**D**): bar graphs showing the ratio of cf-mtDNA-85/p-tau from each group, including all subjects of the entire cohort in (**A**) and from the groups of the AD biomarker-selected cohort in **(D).** ND-Ctrl, patients with neurological diseases and with no clinical signs of AD type dementia; spAD, patients with clinical signs of AD type dementia of slow progression; rpAD: patients with clinical signs of AD of rapid progression. Within squares is the number of patients in each group. Dots represent individual values. Bars represent mean ± 95% CI. BM- = ND-Ctrl patients without AD biomarkers in CSF. BM+ = spAD or AD patients with both low Aβ and high t-tau levels in CSF. (**B**–**F**) Receiving operating curve analyses (ROC) of the whole cohort (**B**,**C**) and the biomarker-selected cohort (**D**) and (**F**). Using the cut-off value of < 0.885, the sensitivity and specificity of cf-mtDNA-85/p-tau ratio are higher for patients with spAD in the biomarker-selected cohort (**F**). *** Statistically significantly different, *p* < 0.001; ** statistically significantly different, *p* < 0.01; * statistically significantly different *p* < 0.05; n.s = nonsignificantly different.

**Table 1 ijms-21-06298-t001:** Cerebrospinal fluid (CSF) levels and Alzheimer’s disease (AD) Biomarkers in patients and neurological disease controls.

	*n*	Gender (f/m)	Age	cf-mtDNA-85 (copies/uL CSF)	t-tau (pg/mL CSF)	Ab1-42 (pg/mL CSF)	p-tau (pg/mL CSF)
ND-Controls	49	26/23	69 (66,72)	62 (47,77)	323 (260,387)	646 (564,729)	46 (38,53)
Absence of AD Biomarker (Ab1-42 > 450 & t-tau < 450 (pg/mL))	31	16/15	71 (67,74)	69 (48,91)	243 (210,277)	772 (680,864)	37 (33,42)
Presence AD Biomarker (Ab1-42 ≤ 450 or t-tau ≥ 450 (pg/mL)	18	10/8	66 (62,70)	49 (33,65)	460 (309,612) *	430 (327,533) *	61 (42,79) *
AD	46	25/21	68 (65,71)	43 (31,55) #	649 ± (538,759) #	444 (389,499) #	99 (84,113) #
Presence of AD Biomarkers (Ab1-42 ≤ 450 & t-tau ≥ 450 (pg/mL))	23	13/10	66 (62,70)	35 (19,51)	846 (680,1012) *	338 (306,370) *	117 (91,143) *
spAD	14	7/7	66 (60,73)	22 (15,29)	753 (572,934)	336 (297,375)	106 (71,142)
rpAD	9	6/3	65 (60,70)	55 (14,96)	991 (642,1340)	341 (272,410)	133 (88,177)
Absence of AD Biomarker (Ab1-42 > 450 or t-tau < 450 (pg/mL)	23	12/11	70 (66,74)	51 (33,69)	451 (349,553)	550 (463,638)	81 (70,92)
spAD	16	9/7	69 (64,74)	46 (26,66)	476 (346,606)	518 (418,619)	85 (73,97)
rpAD	7	3/4	71 (62,80)	61 (12,111)	394 (187,602)	623 (412,835)	71 (40,102)

The study cohort consisted of a total of 95 subjects recruited at the Clinical Dementia Center Göttingen (Germany) and classified in two groups according to clinical diagnosis: patients diagnosed with neurological diseases without dementia (ND, *n* = 49) and Alzheimer’s disease (AD, *n* = 46). Both groups were sub-classified according to the presence or absence of AD Biomarkers (Aβ1-42 > 450 & t-tau < 450 (pg/mL)). The AD group was further stratified in slow progressive AD (sAD) and rapidly progressive AD (rpAD). Values are mean ± 95% CI. * Significantly different from the absence of AD biomarker group. # significantly different from ND-Controls. All *p* < 0.05. No significant differences were observed in age and gender between groups.

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
