# Peer review of "Cerebrospinal Fluid Mitochondrial DNA in Rapid and Slow Progressive Forms of Alzheimer’s Disease"

_ijms, 2020, doi:10.3390/ijms21176298_

Round 1
Reviewer 1 Report
Podlesniy et al. analyse in their article “Cerebrospinal fluid mitochondrial DNA in rapid and slow progressive forms of Alzheimer’s disease” the relationship between cf-mtDNA and other biomarkers of Alzheimer’s type dementia. Their main finding is that cf-mtDNA in CSF correlates with the markers Aβ and p-tau, but not with the marker of neuronal damage t-tau. The authors hypothesize that low cf-mtDNA together with low Aβ and high p-tau constitute a distinctive CSF biomarker profile.
The paper is relatively easy to follow, and the authors acknowledge a limitation of their study, i.e. the lack of statistically significant difference between spAD and rpAD groups. The data presented show a robust differentiation between the control groups and spAD and rpAD groups.
Major points
The dPCR protocol lacks important information as e.g. the temperature protocol. I would advise to apply the dMIQE guidelines (Huggett et al., 2013). Moreover, as no DNA extraction was applied, maybe a spike-in of DNA could have ensured that the results are not influenced by e.g. inhibitors that vary between patients (though digital PCR is relatively robust). This should either be discussed or experimentally proven.
Minor points
In the Figure 3 legend, first sentence, “groups of the AD biomarker-selected cohort in B)” should either read “D” or is to me confusing.
I find it confusing that the statistical values are given directly, maybe the limits for the number of asterisks can additionally be defined.
“slow progression (spAD)” vs “sp-AD” should be consistent and only one time explained
Huggett, J.F., Foy, C.A., Benes, V., Emslie, K., Garson, J.A., Haynes, R., Hellemans, J., Kubista, M., Mueller, R.D., and Nolan, T. (2013). The Digital MIQE Guidelines: M inimum I nformation for Publication of Q uantitative Digital PCR E xperiments. Clin Chem 59, 892-902.
Reviewer 2 Report
In this study Authors measured levels of cell-free mitochondrial DNA (cf-mtDNA) and biomarkers of Alzheimer’s disease (β-amyloid, total tau e p-tau) in cerebrospinal fluid (CSF) of 46 subjects with clinical diagnosis of AD and of 49 subjects with neurological diseases without dementia (ND), that served as controls. Authors failed to show that cf-mtDNA levels may distinguish between AD patients with slow and rapid progression of the disease. Authors speculated that low cf-mtDNA levels may identify initial phases of the disease based on correlation of cf-mtDNA with Ab and phospho-tau but not with total tau. However, Authors did not measure cf-mtDNA levels in prodromal AD patients. Thus, they did not prove their hypothesis. Other limitations of the study are the following.
- Authors should describe the specific neurological diagnoses of the 49 control patients.
- Half of the 46 patients with diagnosis AD did not present positivity for both AD CSF biomarkers (Ab42 ≤ 450 pg/mL & t-tau ≥ 450 pg/mL). This low percentage is quite worrying and rises questions of differential diagnosis of dementia.
- Both AD and control groups were sub-classified according to the positivity or negativity for AD CSF biomarkers (Aβ42 ≤ 450 pg/mL & t-tau ≥ 450 pg/mL). The AD group was further stratified in slow (spAD) and fast (rpAD) progressor. The result of these subclassification is that we have groups with sample size ranging from 7 to 31 subjects. I wonder if the limited sample size of this study may allow to make robust conclusions. If we consider the group of patients with “robust” AD diagnosis, i.e., those with positivity for AD CSF biomarkes (n=23), there were only 14 patients with slow decline and 9 patients with fast decline. These two groups presented quite overlapping 95% confidence intervals of cf-mtDNA (15-29 vs 19-96 copies/mL, respectively, Table 1). There was a high interindividual variability in CSF cf-mtDNA content between AD subjects and controls (Figure 1A), between slow and fast declining AD patients (Figure 1B) and between CSF AD positive and negative subjects (Figure 2).
- Apparently there is inconsistency between Table 1 and Figure 1B on the statistical significance in the differences in mean CSF cell-free mitochondrial DNA content between slow and rapid decliners within the AD group. According to Table 1 the difference is statistically significant but according Figure 1B it is not significant.
- I do not think that comparing mean CSF cell-free mitochondrial DNA content between AD CSF biomarkers groups irrespectively from clinical diagnosis is meaningful (Figure 2).
- In the Introduction, Authors reported the definition of rapid progressive AD (rpAD) and supported it with a number of references (lines 47-54). However, Authors did not define slow progressive AD (spAD) and did not report relevant references.
- In the Results section, there is no need repeating all data of AD CSF biomarkers shown in Table 1.
Reviewer 3 Report
Podlesniy et. al. investigated the relationship between cf-mtDNA and classic biomarkers of AD in slow and rapid progression AD compared to ND-Ctrl patients. The manuscript was well written and easy to follow. Their conclusions and interpretation of the data was appropriate for the results that were displayed. I had a few minor edits.
1) Please explain in greater detail in section 2.2 Line 110 what types of patients ND-controls are? Currently, it appears that they are simply healthy individuals with no AD-type dementia...and later you reveal that they have a neurological disorder ( a number of different types) but lack AD-type dementia.
2) Figure 2 B, D, F...please add (pg/mL) to x-axis to be consistent.
3) Why was a cutoff of .885 used in Figure 3?
4) Additional elaboration/explanation of ROC analysis in methods may be useful for readers that are unfamiliar with this type of analysis.
Thank you.
Round 2
Reviewer 2 Report
I think the revised manuscript is significantly improved